# Deep Learning-Based Detection of Aflatoxin B1 Contamination in Almonds Using Hyperspectral Imaging: A Focus on Optimized 3D Inception–ResNet Model

**DOI:** 10.3390/toxins17040156

**Published:** 2025-03-22

**Authors:** Md. Ahasan Kabir, Ivan Lee, Sang-Heon Lee

**Affiliations:** 1UniSA STEM, University of South Australia, Mawson Lakes, Adelaide, SA 5095, Australia; ivan.lee@unisa.edu.au (I.L.); sang-heon.lee@unisa.edu.au (S.-H.L.); 2Department of Electronics and Telecommunication Engineering, Chittagong University of Engineering and Technology, Chittagong 4349, Bangladesh

**Keywords:** aflatoxin B1, hyperspectral imaging, Inception–ResNet, convolutional neural network, deep learning, AUC, ResNet, feature selection

## Abstract

Aflatoxin B1, a toxic carcinogen frequently contaminating almonds, nuts, and food products, poses significant health risks. Therefore, a rapid and non-destructive detection method is crucial to detect aflatoxin B1-contaminated almonds to ensure food safety. This study introduces a novel deep learning approach utilizing 3D Inception–ResNet architecture with fine-tuning to classify aflatoxin B1-contaminated almonds using hyperspectral images. The proposed model achieved higher classification accuracy than traditional methods, such as support vector machine (SVM), random forest (RF), quadratic discriminant analysis (QDA), and decision tree (DT), for classifying aflatoxin B1 contaminated almonds. A feature selection algorithm was employed to enhance processing efficiency and reduce spectral dimensionality while maintaining high classification accuracy. Experimental results demonstrate that the proposed 3D Inception–ResNet (Lightweight) model achieves superior classification performance with a 90.81% validation accuracy, an F1-score of 0.899, and an area under the curve value of 0.964, outperforming traditional machine learning approaches. The Lightweight 3D Inception–ResNet model, with 381 layers, offers a computationally efficient alternative suitable for real-time industrial applications. These research findings highlight the potential of hyperspectral imaging combined with deep learning for aflatoxin B1 detection in almonds with higher accuracy. This approach supports the development of real-time automated screening systems for food safety, reducing contamination-related risks in almonds.

## 1. Introduction

Almonds are one of the vital agricultural products, recognized for their exceptional nutritional value, economic significance, and global demand [1]. In 2023, global almond production reached approximately 1.7 million metric tons, with the United States (US) remaining the largest producer, followed by Australia, and the European Union (EU) [2]. Almonds are used in various food products, including almond milk, confectioneries, and healthy snacks, underscoring the need for strict quality controls during production and processing [1]. Despite their significance, almonds are highly susceptible to aflatoxin B1 (AFB1) contamination, primarily caused by *Aspergillus flavus* and *Aspergillus parasiticus* fungi, which thrive under warm and humid conditions, infecting almonds both pre- and post-harvest [3]. The prevalence of AFB1 contamination has been further exacerbated by global climate change, as the increasing frequency of extreme weather events creates favorable conditions for fungal growth and toxin production [4,5]. AFB1 is the most potent and hazardous aflatoxin and has been classified as a group 1 carcinogen by the International Agency for Research on Cancer (IARC). The health impacts of AFB1 contamination are severe and wide-ranging. Long-term exposure is closely associated with hepatocellular carcinoma, a major contributor to cancer-related mortality globally [6,7].

Beyond the health implications, the economic consequences are equally significant, as approximately two-thirds of rejected export shipments to European countries are attributed to unsafe levels of aflatoxins [8]. Hence, almonds are subject to strict regulatory limits imposed by organizations such as the United States Food and Drug Administration (FDA) and the European Food Safety Authority (EFSA). These limits ensure that products entering the market meet food safety standards, protecting consumers from harmful exposure [9]. Non-compliance due to contamination can lead to significant financial losses for producers, including rejected shipments, product recalls, and reputational damage [10]. For small-scale farmers, such losses can be devastating, exacerbating economic inequalities in agricultural communities. The convergence of health and financial risks underscores the urgent need for effective AFB1 detection and mitigation strategies, making rapid, reliable, and non-destructive detection methods essential for the almond industry. Conventional techniques for AFB1 detection, such as enzyme-linked immunosorbent assay (ELISA) [11] and high-performance liquid chromatography (HPLC) [12], provide accurate quantification, but are labor-intensive, expensive, and time-consuming. Moreover, these methods are destructive, unsuitable for large-scale industrial applications, and require skilled personnel [13]. Alternative spectroscopic methods, such as fluorescence and near-infrared spectroscopy, provide non-destructive testing but struggle with sensitivity and specificity in complex matrices like almonds, where variations in composition and surface properties can interfere with accurate measurements [14,15].

Hyperspectral imaging (HSI) integrates the capabilities of imaging and spectroscopy to capture spatial and spectral information concurrently. By capturing reflectance or fluorescence spectra across hundreds of wavelengths, HSI enables differentiation between contaminated and uncontaminated samples based on their distinct spectral signatures [16,17]. Additionally, HSI has shown remarkable success in identifying and quantifying mycotoxins in various agricultural products. Numerous studies have demonstrated the efficacy of HSI for mycotoxins detection in cereals and nuts, leveraging its capability to analyze chemical composition non-destructively. For instance, HSI has been used to detect AFB1 in maize kernels, achieving high classification accuracy by utilizing near-infrared and short-wave infrared spectra [5,9]. Wu et al. applied HSI to detect toxigenic fungi and aflatoxins in various nuts, including almonds, using chemometric models for data interpretation [4]. Torres-Rodríguez et al. highlighted the utility of HSI in distinguishing bitter almonds from sweet varieties, showcasing its versatility in nut classification tasks [14]. These efforts underlined the potential of HSI as a non-invasive, accurate, and scalable method for aflatoxin detection across diverse commodities. Zhu et al. developed a deep convolutional neural network (CNN) designed to extract spectral–spatial features from hyperspectral data, enhancing the detection accuracy of AFB1 in peanuts [9]. This approach addressed the challenges of high-dimensional data processing, demonstrating its potential for real-time industrial applications.

Research focused on almonds has gained momentum in recent years, aiming to overcome the limitations of traditional methods. Mishra et al. employed a multispectral imaging system to detect AFB1 in single-kernel almonds, achieving promising classification results through chemometric analysis [10]. Kabir et al. developed a Gaussian process and support vector regression framework for quantifying AFB1 contamination levels in almonds, utilizing HSI data in the short-wave infrared range [17]. Their research investigated selectively chosen almonds of identical shape and thickness with consistent HSI response.

Despite significant advancements in hyperspectral imaging for AFB1 detection, several critical challenges remain unresolved, particularly in the context of almonds. Additionally, conventional methods, such as support vector machines (SVM) and quadratic discriminant analysis (QDA) are widely used in classification tasks and face inherent limitations, including sensitivity to noise [18,19] and the inability to effectively capture the intricate spectral–spatial relationships within HSI data. The high dimensionality and spectral correlation inherent in HSI exacerbate these issues, often resulting in suboptimal classification performance. Additionally, the thickness and texture variation of almond kernels and their complex spectral responses introduce further variability, which traditional machine-learning approaches are ill-equipped to handle. While 2D deep convolutional neural networks (DCNNs) have been explored in related contexts, their architecture is not designed to manage the volumetric nature of HSI data, leading to inefficiencies in capturing inter-band spectral correlations.

This work addresses these research gaps by proposing a 3D deep learning model for high-dimensional HSI data. The research contributions include the following: (1) developing a lightweight 3D Inception–ResNet by modifying the kernel size and convolutional parameters to enhance efficiency; (2) leveraging 3D convolutions for spectral–spatial feature extraction, addressing the limitations of 2D CNNs; (3) incorporating Inception modules for multi-scale learning and residual connections for improved gradient flow; (4) demonstrating superior classification performance compared to traditional machine learning models and other deep learning architecture; and (5) optimizing the model for real-time, inline AFB1 detection in almonds. These improvements ensure better trade-offs between accuracy and computational efficiency.

## 2. Materials and Dataset Preparation

### 2.1. Almond Sample Selection and Preparation

Fresh almonds of the Nonpareil variety were procured from a commercial processing facility in Renmark, South Australia, from the 2022–2023 harvest season. Upon receipt, the almonds were stored at 4 °C in sealed, food-grade zip-lock bags to preserve their freshness until experimental use. Acquiring naturally contaminated almonds through non-destructive methods is unfeasible due to the inability to verify toxin levels without compromising sample integrity. To overcome this limitation, almonds were artificially contaminated under controlled laboratory conditions using standardized AFB1 solutions, ensuring precise and reproducible toxin concentrations for analysis [9,20,21]. An amount of 5 mg AFB1 was purchased from Sigma-Aldrich, a division of Merck Life Science Pty Ltd., located on Mountain Highway, Bayswater, Victoria, Australia. The standard solutions were prepared by diluting the AFB1 standard with a methanol–water mixture (50:50 *v*/*v*). Each almond kernel was standardized to 1 g for consistency and inoculated with 20 µL of AFB1 solution. This resulted in contamination levels of 0.25, 0.5, 0.75, and 1.00 µg/g, corresponding to AFB1 concentrations of 250, 500, 750, and 1000 ppb, respectively. Following contamination, the almonds were air-dried for 48 h in a fume hood to ensure complete solvent evaporation of the methanol–water mixture. Elevated AFB1 concentrations were intentionally used in this study due to the low natural contamination rate (0.03%) reported in commercially contaminated almond batches [22]. This implies that only highly contaminated kernels significantly impact the test outcomes. Also, high-concentration contamination simplifies detection by producing distinct spectral signatures, enabling reliable identification even with limited spectral bands [10,23]. This approach optimizes detection efficiency while reflecting real-world scenarios where removing heavily contaminated almonds is critical for food safety. Control samples (0 µg/g AFB1) were included to establish baseline data. In this way, 5400 almond kernels were prepared for HSI and subsequent analysis.

### 2.2. Hyperspectral Image Acquisition

Hyperspectral imaging was performed using a short-wave infrared (SWIR) imaging system. The imaging setup included an InGaAs detector (Specim FX17e, Oulu, Finland), halogen light sources with optical filters, a motorized conveyor belt for sample transport, and a dedicated computer with a Lumo scanner. The camera operating spectral range of 900 to 1700 nm captures spatially resolved spectral data with a spectral resolution of 8 nm (224 bands), enabling precise detection of molecular absorption features associated with organic compounds, including AFB1. The camera integrates a 640-pixel spatial resolution line-scan sensor and a high-speed imaging capability (up to 330 Hz), making it suitable for real-time or near-real-time food safety inspections. Its cooled InGaAs detector ensures high signal-to-noise ratio (SNR) and thermal stability, critical for consistent measurements in varying environmental conditions. The lighting source employed a 150-watt halogen illumination source paired with a fiber-optic MI-150 high-intensity illuminator (Dolan-Jenner Industries Inc., United States to ensure uniform spectral coverage across the camera’s field of view. Individual almond kernels were placed on the conveyor and scanned at a belt speed of 15 mm/s to ensure consistent image quality. Each hyperspectral data cube comprised 224 spectral bands, capturing the chemical and spatial properties of each kernel. To cover the entire almond, 200 lines were scanned and the acquired HSI image has a dimension of 200×640×224. The captured data cubes were stored for further spectral analysis.

### 2.3. HPLC Analysis for Sample Reference

High-performance liquid chromatography was used to quantify the concentrations of AFB1 in contaminated almonds, ensuring the accuracy and consistency of the contamination process and its reliability for experimental assessments. Eight almonds (to fit in a 50 mL vial tube) were ground to a fine powder and mixed with 40 mL of methanol–water (50:50) solution. The mixtures were shaken for an hour and left to settle. The slurry was filtered through 0.45 µm disc filters, and the extract was collected for the analysis. An Agilent 1200 series HPLC system, integrated with a fluorescence detector, was used to measure AFB1 contamination levels. The system operated with a reverse-phase Zorbax ODS column and a mobile phase of water, acetonitrile, and methanol (1:1:3), delivered at a flow rate of 1 mL/min. To establish reliable ground-truth labels for model development, in this research 40 almond samples were prepared for HPLC analysis, with 8 replicates per contamination level 0, 0.25, 0.5, 0.75, and 1.00 µg/g, respectively. In addition, to independently validate the spiking process, three independent batches (20 almonds per batch) were analyzed by the National Measurement Institute (NMI, Australia), a certified reference laboratory. Comparative HPLC results from both in-house and NMI tests demonstrated strong alignment (R^2^ = 0.98, *p* < 0.01), confirming the precision of the laboratory-controlled contamination method used in this research. Therefore, the artificially AFB1-contaminated concentrations were used as benchmark data to train and evaluate machine learning models for classifying hyperspectral images.

### 2.4. Hyperspectral Image Processing

Hyperspectral imaging systems inherently exhibit thermal noise due to dark current, which persists even when the sensor is shielded from light. This noise, influenced by ambient temperature fluctuations, introduces artifacts into the spectral data [24]. To mitigate this, radiometric calibration was performed using a two-step correction protocol. First, a dark reference image ($I_{Dark}$) was captured with the camera shutter closed to quantify baseline noise. Next, a white reference image ($I_{White}$) was acquired using a polytetrafluoroethylene calibration panel under uniform illumination. The raw hyperspectral image ($I_{Raw}$) was then normalized using Equation (1) [25]:(1)$$I=\left(I_{Raw}-I_{Dark}\right)/\left(I_{White}-I_{Dark}\right)$$

For spatial segmentation of almond kernels, an RGB composite image derived from hyperspectral bands (1170 nm, 1173 nm, and 1156 nm) was thresholded with minor adjustments to isolate almond pixels from the background. The resulting binary mask was applied across all spectral bands in the hypercube, enabling precise extraction of almond regions while excluding non-target artifacts. This method minimized spectral contamination from background elements, ensuring robust region-of-interest (ROI) analysis. The segmented ROI was resized to 150×100×224 dimensions to suit the deep learning model input, resulting in a total of 5400 HSI images. However, to balance the number of samples between the contaminated and non-contaminated classes for model development, 1804 samples were randomly removed from the higher contamination levels (500, 750, and 1000 ppb), reducing the dataset to 3596. The final dataset comprised 1798 samples of 0 ppb, 892 samples of 250 ppb, 706 samples of 500 ppb, 100 samples of 750 ppb, and 100 samples of 1000 ppb contamination levels. Higher contamination levels (e.g., 750 ppb and 1000 ppb) typically exhibit stronger spectral signatures than lower contamination levels (e.g., 250 ppb and 500 ppb). Therefore, the use of an uneven distribution of contamination levels was necessary to ensure the model could effectively detect AFB1 contamination across a range of concentrations.

### 2.5. Data Dimensionality Reduction

In this study, we evaluated the developed model in two ways. First, the model was trained using 10 principal components (PCs), which are equivalent to utilizing the full spectra. Second, the model was trained based on a specific number of spectral bands (4, 6, and 8) selected using a feature selection algorithm. The purpose of this approach is to develop a model suitable for industrial inline applications, ensuring high accuracy while minimizing computational costs. Principal component analysis (PCA) was employed to reduce the dimensionality of the hyperspectral data by transforming correlated features into uncorrelated principal components that capture the maximum variance. PCA was applied to a data cube of size 200 × 640 × 224. This technique reduces the dimensionality of hyperspectral data by projecting it onto orthogonal axes, called principal components, which are ranked based on the variance they capture [26]. In this study, the first 10 PCs were used as an equivalent to the full-spectrum model development, collectively explaining 99.98% of the total variance in the hyperspectral data.

On the other hand, to select a specific number of important features, the correlation-awareness evolutionary sparse hybrid spectral band selection (CAES-HBS) algorithm was introduced [27]. The CAES-HBS algorithm is designed to optimize spectral feature selection for AFB1 classification in hyperspectral imaging. It integrates multiple feature selection methods, including multilayer perceptron and ensemble boosting-based approaches, to identify the most relevant spectral bands across different dimensions. The algorithm employs six boosting ensemble learners alongside decision trees to enhance spectral reliability. A genetic algorithm-based correlation-aware selection process further refines the spectral subset, ensuring minimal redundancy and maximizing classification accuracy. By eliminating highly correlated spectral bands, the CAES-HBS algorithm improves the efficiency and robustness of machine learning models while maintaining high classification performance.

## 3. Research Methodology

This research aims to develop a deep learning classifier with a 3D Inception–ResNet model to detect AFB1 in almonds using hyperspectral images, with the framework illustrated in Figure 1. The proposed model integrates key features of ResNet and Inception networks to train and classify AFB1 contaminated almonds from uncontaminated ones. By building upon the strengths of its 2D predecessors, the model adopts a hybrid architecture that enhances feature extraction and classification accuracy, making it well-suited for hyperspectral image analysis.

### 3.1. 3D ResNet Architecture

The developed 3D ResNet architecture is a specialized deep learning model designed to classify AFB1 contaminated almonds using hyperspectral imaging data. As shown in Figure 2, this architecture processes 3D image cubes through layers of 3D convolution, normalization, pooling, and classification, ensuring robust feature extraction while maintaining computational efficiency. A 3D input layer processes hyperspectral image cubes X with dimension [H×W×D×C], where H, W, D are height, width, and depth, and C (equal 1) is the number of channels. In feature extraction, the initial layer employs a 3D convolutional operation [3×3×3] with a stride of [2×2×2] for extracting high-dimensional spatial–spectral features. The 3D convolution followed by batch normalization and ReLU activation function is mathematically expressed as follows [28]:(2)$$y_{i,j,k}=\sigma \left(\sum_{m,n,p}X_{i+m,\;j+n,\;k+p}.W_{m,n,p}+b\right)/\sqrt{\sigma^{2}+\epsilon }$$
where W represents the kernel weights, b is the bias, σ denotes the ReLU activation, and ϵ is a stabilization constant for batch normalization. The use of 3D convolutions allows simultaneous processing of spatial and spectral dimensions, preserving hyperspectral integrity and overcoming the limitations of 2D models that lose critical spectral information.

#### 3.1.1. Feature Extraction Through Residual Blocks

A series of residual blocks, each incorporating bottleneck designs with three sequential 3D convolutional layers, capture hierarchical features while maintaining computational efficiency. Each block includes the following:A [1×1×1] convolution to reduce dimensionality.A [3×3×3] convolution to extract features.A [1×1×1] convolution to restore dimensionality.

Residual connections skip over these transformations as follows [29]:(3)$$y_{residual}=x+F(x)$$
where F(x) represents a function that is composed of multiple convolutional layers to extract hierarchical features from the input data x. Skip connections mitigate vanishing gradient issues, ensuring efficient gradient propagation across each layer. The model employs stride convolutions and max-pooling layers to handle high-dimensional input data efficiently. The stride convolutions and pooling ensure computational efficiency without significant information loss.

#### 3.1.2. Classification Layers

The classification process begins with a global average pooling layer, which aggregates spatial–spectral information into a compact representation.(4)$$\hat{y}=\frac{1}{N}\sum_{i=1}^{N}x_{i}$$
where N is the total number of spatial elements. This pooled representation is then passed through a fully connected layer, which maps the features to binary classes (presence or absence of AFB1). A softmax activation function computes the final probabilities as follows:(5)$$P\left(y=c|x\right)=exp(z_{c})/\sum_{i}exp(z_{i})$$
where $z_{c}$ is the score for the class c.

### 3.2. 3D Inception Architecture

The 3D Inception architecture builds on the success of the original 2D Inception framework by extending its functionality to volumetric data, making it highly suitable for hyperspectral image classification tasks. The proposed model is explicitly designed to detect AFB1 contamination in almonds by leveraging both spatial and spectral information encoded in HSIs. The 3D adaptation enables multi-scale feature extraction across spatial and spectral dimensions, capturing subtle variations that may indicate contamination while maintaining computational efficiency.

The input hyperspectral data standardizes the scale across spectral bands and is fed into the initial feature extraction layer. Initial feature extraction is performed using a 3D convolutional layer with a kernel size of [3×3×3] followed by a kernel size of [3×3×1], [1×1×1], and a stride of [2×2×2], effectively downsampling the data while capturing low-level spatial–spectral features. This convolutional operation processes the spatial and spectral dimensions simultaneously, preserving critical information essential for identifying chemical differences in contaminated and non-contaminated regions. Batch normalization stabilizes feature distributions, while ReLU activation introduces non-linearity, enhancing the model’s capacity to capture complex spectral relationships.

The 3D Inception blocks are the core of this architecture, designed to extract features at multiple scales simultaneously. The architecture of the developed 3D inception network is illustrated in Figure 3. The network has three different inception blocks (Inception blocks A, B, and C) and two different reduction blocks. The Inception block A contains four parallel branches, each specialized for a distinct type of feature extraction. The branches are as follows:Branch 1 and 2: A [1 × 1 × 1] convolution reduces the input dimensions, lowering computational complexity while retaining essential features for subsequent processing.Branch 3: A [1 × 1 × 1] convolution followed by a [5 × 5 × 5] convolution captures large-scale spatial and spectral patterns.Branch 4: Similar to Branch 2, but with two [3 × 3 × 3] convolutions for detecting immediate-scale patterns that are critical for understanding global spectral variations.

The output of these branches are concatenated along the depth dimension and can be represented as follows:(6)$$Output=F_{1\times 1\times 1}\left(x\right)\left\|F_{1\times 1\times 1}\left(F_{pool}\left(x\right)\right)\right.\left\|F_{5\times 5\times 5}\left(F_{1\times 1\times 1}\left(x\right)\right)\right.\left\|F_{3\times 3\times 3}\left(F_{3\times 3\times 3}\left(F_{1\times 1\times 1}\left(x\right)\right)\right)\right.$$
where $F_{1\times 1\times 1}$, $F_{3\times 3\times 3}$, $F_{5\times 5\times 5}$, and $F_{pool}$ represent the operations in each parallel path, and || is the concatenation operation. By leveraging the multi-scale capabilities of the Inception block, the model processes hyperspectral data with greater precision than traditional architecture. Subtle spectral changes, often localized and sparse, are captured effectively, making this approach highly applicable for detecting AFB1 contamination in almonds.

To handle the computational demands of hyperspectral data, a [3 × 3 × 1] max-pooling operation is adopted to reduce the spatial dimensions while retaining spectral details. Dimensionality reduction in 3D Inception is further optimized using [1 × 1 × 1] convolutions. These bottleneck layers reduce the number of parameters while maintaining the richness of the feature representations, and these scalable deep networks are critical for processing complex hyperspectral datasets, such as those used in high-precision aflatoxin detection. Compared to the 2D Inception architecture, the proposed approach explicitly addresses the additional spectral dimension. The 3D adaptation integrates the spectral dimension directly into the feature extraction pipeline. The use of [3 × 3 × 3] and [5 × 5 × 5] convolutions in Branches 2 and 3 ensures that intermediate and large-scale patterns are detected across both spatial and spectral domains. Additionally, the inclusion of a dedicated pooling branch (Branch 4) adds sensitivity to global trends in spectral data, which is crucial for distinguishing contaminated regions. Despite that, to improve computational cost and gradient flow in Inception blocks B and C, different-sized kernels of [7×1×1] and [1×7×1] were used.

### 3.3. 3D Inception–ResNet Model Architecture

The proposed 3D Inception–ResNet architecture integrates the strengths of 3D ResNet and 3D Inception, creating a hybrid framework that combines residual learning’s efficient gradient propagation with Inception’s multi-scale feature extraction capabilities. The architecture of the proposed 3D Inception–ResNet is illustrated in Figure 4. At the heart of this integration are Inception–ResNet blocks, where multi-branch feature extraction is seamlessly merged with residual connections. Each block consists of parallel branches that extract features at various scales: [1×1×1] convolutions for dimensionality reduction, [3×3×1], [3×3×3], and [5×5×1] convolutions for medium- and large-scale patterns, and average pooling for global context. The outputs of these branches are concatenated along the depth dimension, forming a rich and diverse feature representation. The concatenation of Inception–ResNet block A is as follows:(7)$$y_{mixed}=F_{1\times 1\times 1}\left(x\right)\left\|F_{3\times 3\times 3}\left(F_{1\times 1\times 1}\left(x\right)\right)\right.\left\|F_{3\times 3\times 3}\left(F_{3\times 3\times 3}\left(F_{1\times 1\times 1}\left(x\right)\right)\right)\right.$$

To ensure training stability and enhance gradient flow, the concatenated output from the Inception branches is combined with the input through a residual connection, defined as follows:(8)$$y_{residual}=\alpha .y_{mixed}+x$$
where α is a scaling factor that adjusts the contribution of the Inception block relative to the original input. This integration enables the model to leverage deep residual learning for effective training in deep architecture. Additionally, it utilizes multi-scale contextual learning to capture complex spectral and spatial patterns, particularly for detecting subtle anomalies associated with AFB1 contamination.

The proposed 3D Inception–ResNet architecture integrated features from 3D ResNet and 3D Inception. The model improves upon 3D ResNet by incorporating multi-scale Inception blocks, which enable it to capture localized and global spatial–spectral patterns critical for detecting subtle contamination signals. For instance, the [3 × 3 × 1] and [5 × 5 × 1] branches extract features at varying scales, addressing the limitations of ResNet’s fixed-kernel operations, which may miss finer or broader spectral anomalies. Additionally, residual connections are integrated into the architecture to enhance gradient flow and stabilize training, enabling the model to scale effectively to deeper layers. Unlike standalone Inception models, this hybrid design resolves gradient degradation issues, allowing for the efficient training of complex architecture.

Specialized branches within the 3D Inception–ResNet blocks are tailored to the unique requirements of hyperspectral data. The [1 × 1 × 1] convolution branch isolates critical spectral features while minimizing noise and reducing computational overhead. The [3 × 3 × 1] branch focuses on detecting localized contamination patterns, such as small patches of AFB1, whereas the [5 × 5 × 1] branch identifies broader contamination spread. A pooling branch aggregates global spectral context, which is essential for understanding variations across larger regions of the hyperspectral image. The novel integration of these branches ensures a comprehensive spatial–spectral analysis, enabling the detection of even subtle AFB1 contamination. These modifications make the 3D Inception–ResNet uniquely effective for AFB1 detection using hyperspectral imaging tasks. Its ability to combine fine-grained spatial–spectral feature extraction with robust training stability enables precise and scalable detection of AFB1, addressing the inherent complexity of hyperspectral datasets.

### 3.4. Model Evaluation Metrics

To evaluate the developed machine learning and deep learning model’s performance, three key metrics; accuracy (*Acc*), *F1* score, and area under the curve (AUC) are commonly used [30].(9)$$Acc=\left(T_{P}+T_{N}\right)/\left(T_{P}+T_{N}+F_{P}+F_{N}\right)$$(10)$$F_{1}=2*(Precision\times Recall)/(Precision+Recall)$$
where $Precision=T_{p}/(T_{P}+F_{P})$ and $Recall=T_{N}/(T_{N}+F_{N})$. Accuracy measures the overall correct predictions, the $F_{1}$ score balances precision and recall, and the AUC reflects the model’s ability to distinguish between classes across different thresholds. These three metrics provide a well-rounded view of the model’s performance.

## 4. Results and Discussion

To compare the performances of the proposed model, we developed and evaluated four deep learning models, 3D ResNet, 3D Inception, 3D Inception–ResNet (Deep), and 3D Inception–ResNet (Lightweight), to classify AFB1 contamination using a reduced number of spectral bands. The goal was to design an efficient classification system that maintains high accuracy while minimizing computational complexity, making it suitable for industrial inline applications. Figure 5 visually compares some RGB (using spectral bands of 1170 nm, 1173 nm, and 1156 nm, respectively) images of normal and contaminated almonds, but visually distinguishing contaminated from non-contaminated almonds remains challenging. Deep learning might provide better results; however, considering practical inline industrial applications, processing speed is a key performance measure. Therefore, we first explored conventional machine learning classification algorithms, such as SVM [31], decision tree (DT) [32], random forest (RF) [33], and QDA [34], as these could offer alternative solutions for faster industrial applications. SVM can efficiently handle high-dimensional data by using support vectors, which minimizes the impact of the curse of dimensionality. QDA models non-linear class boundaries by estimating class-specific covariance matrices, which is useful for high-dimensional, complex data, such as the HSI response to AFB1 contamination in almonds with distinct class distributions. RF performs implicit feature selection, thereby improving generalization and reducing overfitting. DT selects informative features and handles irrelevant ones by pruning them to avoid overfitting. Table 1 shows the performance of these machine learning algorithms, with an accuracy of around 75%, and QDA achieving the highest accuracy of 82.38%, an F1 score of 0.8, and an AUC of 0.82 when using eight spectral bands. The experimental data was split 80:20 into training and testing for machine learning, and all models were evaluated using MATLAB 2023b on a computer equipped with an Intel Xeon E5-2620 v3 CPU and 32 GB of RAM. Feature extraction for machine learning models was extensive. A total of 55 features were extracted from each spectrum image, incorporating 4 statistical, 16 texture, 16 Gabor, 2 Fourier, 2 discrete cosines transform, 8 discrete wavelet transforms, 5 gradient, and 2 scale-invariant feature transform features. The combination of diverse features provides interpretability and domain-specific insights while enabling traditional machine-learning models to achieve robust performance.

We further experimented with the developed deep learning and machine learning models to classify AFB1 contamination using reduced spectral bands. The goal was to achieve high classification accuracy while minimizing computational complexity to make the model suitable for industrial inline applications. These experiments helped us evaluate the trade-offs between model complexity and performance in real-time processing settings. An important insight from the spectral subset analysis is that 3D deep learning models maintained high accuracy across different spectral configurations. By contrast, machine learning models showed performance degradation with fewer spectral bands. Notably, the 3D Inception–ResNet (Lightweight) model achieved the highest overall accuracy of 90.81%, with an F1 score of 0.899 and an AUC of 0.964 when using eight spectral bands. This confirms the model’s robustness in handling spectral–spatial complexities, reinforcing its suitability for real-world industrial applications.

Table 2 illustrates the comparative performance of the proposed 3D deep learning models with eight spectra. The 3D ResNet model exhibited subpar performance, with a validation accuracy of only 73.41%, an F1 score of 0.686, and an AUC value of 0.778, highlighting its limitations in handling hyperspectral image data. The 3D Inception performed better than ResNet with a validation accuracy of 86.92%, an F1 score of 0.856, and an AUC value of 0.941. By contrast, the 3D Inception–ResNet (Deep) model achieved 90.59% validation accuracy, an F1 score of 0.893, and an AUC of 0.973, demonstrating superior classification performance. All of the deep learning models were trained using the Adam optimizer with an initial learning rate of 0.001, a minibatch size of 64, and L2 regularization, for 100 epochs to achieve the highest classification performance. The final model was selected based on the best validation error.

In addition to the 3D Inception–ResNet with 824 layers, we conducted experiments with a lighter version consisting of 381 layers, which showed slightly improved performance in validation. This version achieved 90.81% accuracy, an F1 score of 0.899, and an AUC of 0.964, making it a highly efficient and robust model for industrial applications. We explored this lighter version to assess the feasibility of implementing the proposed model in real industrial inline operations that demand faster processing speeds. Notably, this architectural analysis reveals a trade-off between model complexity and performance, with the Deep Inception–ResNet model having 824 layers and 55.9 million parameters, compared to the Lightweight version, which has only 381 layers and 28.8 million parameters. Table 2 compares the testing time of 899 samples of four 3D deep learning models. The 3D ResNet required the longest testing time (27.355 s), indicating slow processing. By contrast, 3D Inception, 3D Inception–ResNet (Deep), and 3D Inception–ResNet (Light) showed significantly faster testing times (4.756 s, 7.494 s, and 5.009 s, respectively) suggesting greater computational efficiency and speed. Despite the reduced complexity and greater computational efficiency, the Lightweight model achieves nearly identical classification accuracy, making it well-suited for real-time industrial applications where computational efficiency is crucial.

This study highlights the potential of deep learning and hyperspectral imaging for scalable and reliable food safety screening. The proposed method offers a rapid, non-destructive, and automated solution for industrial AFB1 detection, addressing the limitations of traditional chemical analyses. Its scalability in industrial applications can be enhanced through hardware optimizations, such as employing faster sensors and edge computing, to improve processing speeds. Similarly, Williams et al. demonstrated the effectiveness of deep spectral–spatial feature extraction in aflatoxin B1 detection using hyperspectral imaging, achieving promising accuracy in peanuts [35]. Additionally, Zhu et al. highlighted the advantages of hyperspectral imaging for non-destructive food safety monitoring, reinforcing the potential of our model for real-time industrial applications [36]. While our model achieves high classification accuracy, certain limitations exist. Due to the unavailability of naturally contaminated almonds in a non-destructive manner, the use of artificially contaminated samples may introduce spectral variations, potentially affecting real-world generalization. Factors such as almond variety and environmental conditions could also influence model performance.

## 5. Conclusions

In this study, we developed and evaluated 3D deep learning models to detect AFB1 contamination in almonds using hyperspectral imaging, focusing on an optimized 3D Inception–ResNet architecture. Four models, 3D ResNet, 3D Inception, and 3D Inception–ResNet, including a lightweight version of 3D Inception–ResNet, were designed and compared against traditional machine learning algorithms, including SVM, QDA, RF, and DT, using an extensive set of handcrafted features. The results demonstrated that both versions of the 3D Inception–ResNet outperformed other models, with the lightweight variant achieving superior performance despite having fewer layers and parameters. This model achieved a classification accuracy of 90.81%, an F1-score of 0.899, and an AUC value of 0.964, surpassing traditional machine learning approaches and other developed deep learning architectures. The dataset, comprising artificially contaminated almonds with varying thicknesses, shapes, and textures, ensured robustness and generalizability to real-world scenarios. The efficiency and high performance of the lightweight 3D Inception–ResNet make it particularly suitable for industrial applications, where computational resources and speed are critical. This study showcases the potential of deep learning-based hyperspectral imaging for precise and scalable AFB1 detection. Its application in industrial inspection systems can significantly improve food safety and quality control in the almond industry. Future work will focus on industrial validation, integration with inline sorting systems, and testing on naturally contaminated samples to ensure real-world applicability.

## Figures and Tables

**Figure 1 toxins-17-00156-f001:**
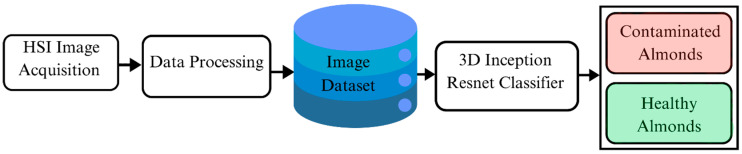
Aflatoxin B1 detection framework.

**Figure 2 toxins-17-00156-f002:**
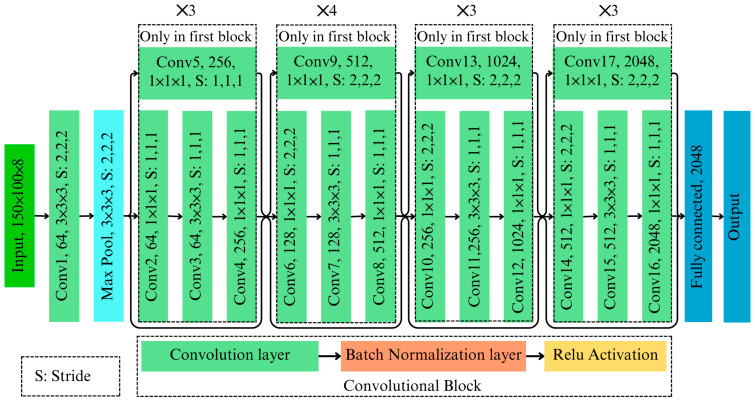
Developed 3D ResNet network architecture.

**Figure 3 toxins-17-00156-f003:**
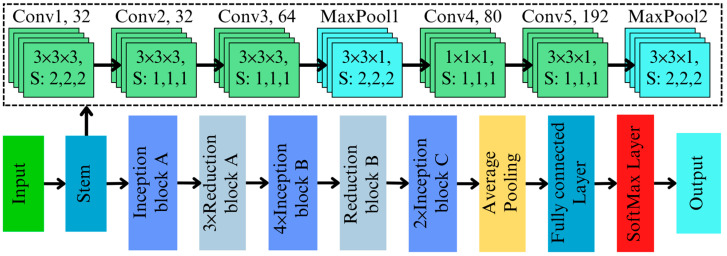
3D Inception network architecture to classify aflatoxin B1 in almonds.

**Figure 4 toxins-17-00156-f004:**
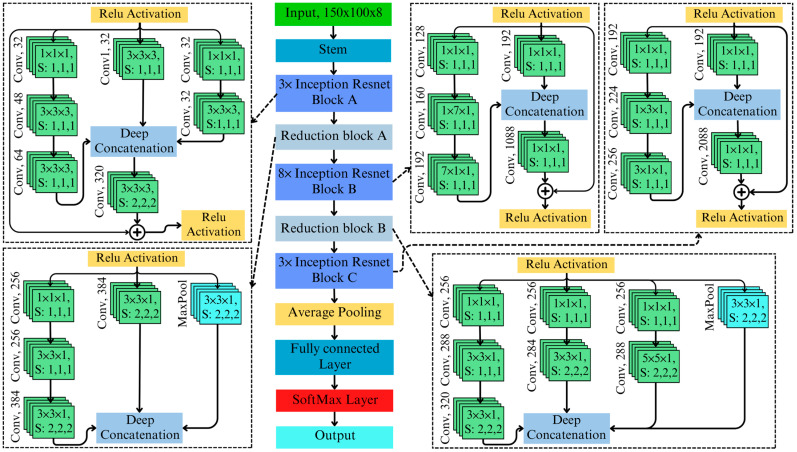
Framework of developed 3D Inception–ResNet model for aflatoxin B1 detection in almonds.

**Figure 5 toxins-17-00156-f005:**
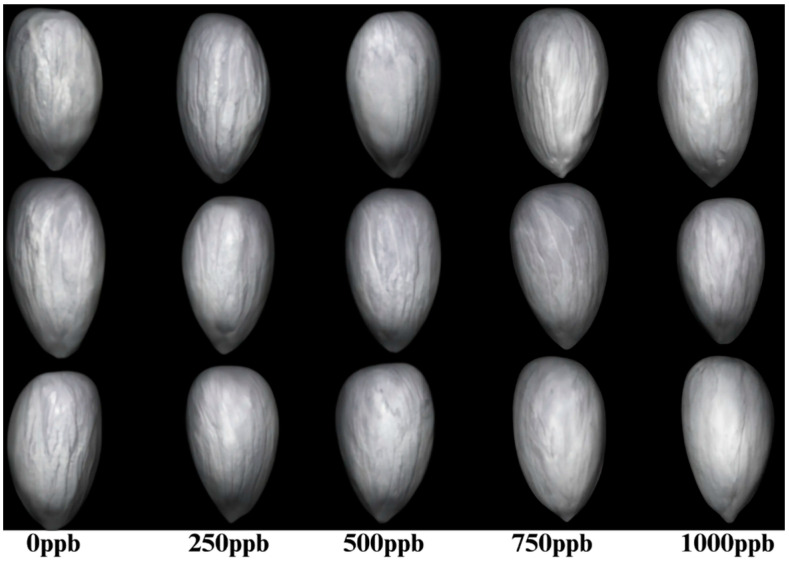
Visualization of normal and contaminated almonds using 3 bands of HSI.

**Table 1 toxins-17-00156-t001:** Comparison of aflatoxin B1 classification performance of classical machine learning and developed 3D CNN model.

Model Name	4 Spectra			6 Spectra			8 Spectra			Full Spectra		
	Acc	F1	AUC	Acc	F1	AUC	Acc	F1	AUC	Acc	F1	AUC
SVM	76.40	0.75	0.76	77.38	0.75	0.77	78.13	0.76	0.78	76.54	0.750	0.765
DT	67.20	0.64	0.67	66.45	0.63	0.66	66.45	0.63	0.66	70.10	0.677	0.699
RF	69.49	0.66	0.69	70.09	0.66	0.70	70.93	0.67	0.71	73.32	0.708	0.731
QDA	77.76	0.76	0.78	80.70	0.79	0.80	82.38	0.80	0.82	76.26	0.743	0.761
3D ResNet	69.3	0.614	0.740	71.03	0.658	0.767	73.41	0.686	0.778	75.89	0.698	0.827
3D Inception	82.70	0.831	0.891	85.44	0.842	0.933	86.92	0.856	0.941	78.70	0.752	0.867
3D Inception–ResNet (Deep)	85.24	**0.863**	0.935	**89.41**	**0.886**	**0.959**	90.59	0.893	**0.973**	86.70	0.86	**0.947**
3D Inception–ResNet (Lightweight)	**86.81**	0.854	**0.937**	87.89	0.867	0.954	**90.81**	**0.899**	0.964	**87.73**	**0.864**	**0.947**

**Table 2 toxins-17-00156-t002:** Performance comparison of different developed 3D deep learning models for 8 significant spectra.

	Layers	Para-Meters	Training			Testing				Validation		
			Acc	F1	AUC	Acc	F1	AUC	Time (s)	Acc	F1	AUC
3D ResNet	349	87.3 M	76.92	0.73	0.843	70.72	0.671	0.777	27.355	73.41	0.686	0.778
3D Inception	315	29.8 M	92.16	0.914	0.979	86.92	0.864	0.956	4.756	86.92	0.856	0.941
3D Inception–ResNet (Deep)	824	55.9 M	97.11	0.978	0.998	91.14	0.904	0.975	7.494	90.59	0.893	0.973
3D Inception–ResNet (Light)	381	28.8 M	96.14	0.960	0.989	90.70	0.904	0.960	5.009	90.81	0.899	0.964

## Data Availability

The datasets presented in this article are not readily available because the data are part of an ongoing study.

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
