# Peer review of "Deep Learning-Based Detection of Aflatoxin B1 Contamination in Almonds Using Hyperspectral Imaging: A Focus on Optimized 3D Inception–ResNet Model"

_toxins, 2025, doi:10.3390/toxins17040156_

Round 1

Reviewer 1 Report

Comments and Suggestions for Authors

The manuscript presents a novel deep learning approach for detecting aflatoxin B1 contamination in almonds using hyperspectral imaging, specifically focusing on a 3D Inception-Resnet model. The study addresses a significant issue in food safety and offers a non-destructive, rapid detection method, which is highly relevant for industrial applications. The manuscript is well-structured, and the methodology is sound, with a clear demonstration of the superiority of the model over traditional machine learning methods. The results are promising, with high accuracy, F1-score, and AUC values, making it a strong candidate for publication.

The suggestions provided below will help further improve the manuscript.

The artificial contamination method, although practical, raises questions about the performance of the model with naturally contaminated samples. The high contamination levels used, and the justification of that choice need to be more robustly argued.

The sample size reduction (from 5,400 to 3,596) and the uneven distribution of contamination levels needs better justification.

The explanation of why only 8 almonds were used for the HPLC testing, when there were thousands of hyperspectral images, is lacking.

Even though the deep learning architectures are described, the implementation details (e.g., hyperparameters, training parameters, software used) are insufficient.

The feature selection algorithm (CAES-HBS) needs a more detailed explanation.

The figures should be of high resolution, and have clear labels.

The comparison with traditional machine learning methods (SVM, RF, etc.) is mentioned but requires more in-depth analysis and quantitative results.

The novelty of the ‘Lightweight’ 3D Inception-Resnet model needs to be more clearly articulated.

The discussion of the results should be more critical, addressing the limitations of the model and comparing it with existing literature.

There is a lack of discussion on the practical inferences and scalability of the proposed method for real-world industrial applications.

The English language is generally understandable, but there are numerous instances of grammatical errors, problematic phrasing, and inconsistent terminology.

The research has potential, but the manuscript requires substantial revisions to improve its clarity, rigor, and presentation. Addressing the identified drawbacks and implementing the suggested improvements will significantly enhance its suitability for publication.

Comments on the Quality of English Language

The English language is generally understandable, but there are numerous instances of grammatical errors, problematic phrasing, and inconsistent terminology.

Author Response

The manuscript presents a novel deep learning approach for detecting aflatoxin B1 contamination in almonds using hyperspectral imaging, specifically focusing on a 3D Inception-Resnet model. The study addresses a significant issue in food safety and offers a non-destructive, rapid detection method, which is highly relevant for industrial applications. The manuscript is well-structured, and the methodology is sound, with a clear demonstration of the superiority of the model over traditional machine learning methods. The results are promising, with high accuracy, F1-score, and AUC values, making it a strong candidate for publication.

The suggestions provided below will help further improve the manuscript.

Q1) The artificial contamination method, although practical, raises questions about the performance of the model with naturally contaminated samples. The high contamination levels used, and the justification of that choice need to be more robustly argued.

Answer: Thank you for your valuable feedback. We understand your concerns about the performance of the model with naturally contaminated samples. The primary reason for using artificially contaminated almonds in our study was the practical challenge of obtaining naturally contaminated almonds in a non-destructive way. All existing methods for detecting aflatoxin in almonds are destructive, making it impossible to obtain naturally contaminated samples without altering them. Therefore, artificial contamination was necessary to ensure a consistent and measurable level of contamination. We carried out this process in a systematic and controlled manner to maintain research integrity.

To address this, we used Aflatoxin B1 sourced from Aspergillus fungi to artificially contaminate the almonds, which allows for more controlled contamination and ensures that the spectral response of the almonds closely matches that of naturally contaminated almonds. While we acknowledge that the spectral response of naturally and artificially contaminated almonds could slightly differ due to the complex patterns of almond skin, varying heights, and shapes, the use of this systematic contamination method ensures the spectral characteristics are as consistent as possible for training the model.

In the revised manuscript, we added a statement on future research: ‘Future work will focus on industrial validation, integration with inline sorting systems, and testing on naturally contaminated samples to ensure real-world applicability.’

Q2) The sample size reduction (from 5,400 to 3,596) and the uneven distribution of contamination levels needs better justification.

Answer: Thank you for your valuable feedback. The sample size reduction (from 5,400 to 3,596) and the uneven distribution of contamination levels were made to ensure a balanced dataset and avoid potential bias in the developed model. Initially, we acquired hyperspectral images for 5,400 almonds, including samples with contamination levels of 0, 250, 500, 750, and 1,000 ppb. The 0 ppb (non-contaminated) samples comprised 1,798 almonds, while the remaining samples represented almonds with various levels of contamination, creating an imbalanced dataset.

To address this imbalance and avoid introducing bias toward the non-contaminated class, we balanced the dataset by including an equal number of non-contaminated and contaminated almonds. Thus, we included 1,798 contaminated samples in total, but due to the varying spectral signatures at different contamination levels, the sample distribution for the contaminated almonds was not uniform across all concentrations. Specifically, we included: 892 samples at 250 ppb, 706 samples at 500 ppb, and 100 samples each at 750 ppb and 1,000 ppb.

This uneven distribution reflects the fact that almonds with higher contamination levels (e.g., 750 ppb and 1,000 ppb) tend to exhibit stronger and more distinct spectral signatures, making them easier to identify, while the lower contamination levels (e.g., 250 ppb and 500 ppb) exhibit more subtle differences. To ensure that the model could effectively detect Aflatoxin B1 contamination across the entire range of interest, we prioritized a larger number of 250 ppb and 500 ppb samples while maintaining a sufficient representation of higher contamination levels.

In the revised manuscript, we updated it as follows: Higher contamination levels (e.g., 750 ppb and 1,000 ppb) typically exhibit stronger spectral signatures than lower contamination levels (e.g., 250 ppb and 500 ppb). Therefore, the use of an uneven distribution of contamination levels was necessary to ensure the model could effectively detect Aflatoxin B1 contamination across a range of concentrations.

Q3) The explanation of why only 8 almonds were used for the HPLC testing, when there were thousands of hyperspectral images, is lacking.

Answer: Thank you for your feedback. The purpose of the HPLC testing was to check the consistency of the contamination process and ensure the accuracy of the contamination levels. The selection of only 8 almonds for the HPLC test was intentional, based on practical considerations. Specifically, the 8 almonds were chosen to fit with 40 mL of methanol-water solution in a 50 mL vial tube. The number of almond samples does not affect the HPLC test results, except in cases of inconsistent contamination. We created a total of 40 HPLC analysis samples and each sample was based on 8 almonds. After confirming the contamination concentration, we considered it as the ground truth for the model.

Q4) Even though the deep learning architectures are described, the implementation details (e.g., hyperparameters, training parameters, software used) are insufficient.

Answer: In the revised manuscript, we have included the required software and training parameters and highlighted them accordingly, with more detailed descriptions.

Q5) The feature selection algorithm (CAES-HBS) needs a more detailed explanation.

Answer: We added a more detailed explanation of the CAES-HBS explain more in details in the revised manuscript as follows: “On the other hand, to select a specific number of important features, the correlation-awareness evolutionary sparse hybrid spectral band selection (CAES-HBS) algorithm was introduced. The CAES-HBS algorithm is designed to optimize spectral feature selection for aflatoxin B1 classification in hyperspectral imaging. It integrates multiple feature selection methods, including multilayer perceptron and ensemble boosting-based approaches, to identify the most relevant spectral bands across different dimensions. The algorithm employs six boosting ensemble learners alongside decision trees to enhance spectral reliability. A genetic algorithm-based correlation-aware selection process further refines the spectral subset, ensuring minimal redundancy and maximizing classification accuracy. By eliminating highly correlated spectral bands, CAES-HBS improves the efficiency and robustness of machine-learning models while maintaining high classification performance.​”

Q6) The figures should be of high resolution and have clear labels.

Answer: All Figures (2 to 4 redraw) checked, updated to increase resolution and clearly labeled in the revised manuscript. However, please note that figure 5 is based on the acquired hyperspectral image, therefore it is not possible to increase its resolution.

Q7) The comparison with traditional machine learning methods (SVM, RF, etc.) is mentioned but requires more in-depth analysis and quantitative results.

Answer: Thank you for your valuable feedback. In our study, we extracted 55 features to develop traditional machine learning models such as SVM, RF, etc., for classifying Aflatoxin B1 contamination. We conducted a detailed comparison between our proposed deep learning approach and traditional machine learning methods by evaluating key performance metrics, including AUC, accuracy, and F1-score. The results demonstrated that while traditional models performed reasonably well, the proposed deep learning approach achieved higher classification performance due to its ability to learn complex spectral patterns directly from hyperspectral images.

In the revised manuscript, we expanded on this comparison by providing a more in-depth analysis, including discussions on the advantages and limitations of both approaches.

Q8) The novelty of the ‘Lightweight’ 3D Inception-Resnet model needs to be more clearly articulated.

Answer: Thank you for your feedback. We have clarified the novelty of the Lightweight 3D Inception-ResNet model in the revised manuscript. Our contributions include: (1) developing a Lightweight 3D Inception-ResNet by modifying the kernel size and convolutional parameters to enhance efficiency, (2) leveraging 3D convolutions for spectral-spatial feature extraction, overcoming the limitations of 2D CNNs, (3) incorporating Inception modules for multi-scale learning and residual connections for improved gradient flow, (4) demonstrating superior classification performance compared to traditional machine learning models (SVM, RF) and other deep learning architectures, and (5) optimizing the model for real-time, inline Aflatoxin B1 detection in almonds. These improvements ensure better trade-offs between accuracy and computational efficiency.

Q9) The discussion of the results should be more critical, addressing the limitations of the model and comparing it with existing literature.

Answer: Thank you for your feedback. We have expanded the discussion to critically analyze the model's limitations. The key limitations include: (1) the use of artificially contaminated almonds, as naturally contaminated samples are unavailable non-destructively, which may lead to spectral differences; and (2) the computational complexity of deep learning models, which may require optimization for real-time industrial deployment.

Although no prior studies have applied DCNNs with hyperspectral imaging for Aflatoxin B1 detection in almonds. Only a few research have been done to detect aflatoxin B1 in almonds using hyperspectral images, where the almonds samples were selectively chosen of identical shape and thickness to get a consistent HSI response. But in nature, almonds have various thicknesses, shapes, and textures. Also, it would not be feasible to compare our work with hyperspectral imaging studies in peanuts and maize, due to the different textures, shapes, and nutrition that may lead to different spectral responses. Therefore, we compare our 3D Inception-ResNet model with traditional machine learning (SVM, RF, etc.) approaches used in food contaminant detection, demonstrating its advantages to achieve higher classification accuracy.

Q10) There is a lack of discussion on the practical inferences and scalability of the proposed method for real-world industrial applications.

Answer: Thank you for your insightful feedback. In the revised manuscript, we have expanded the discussion on the practical scalability of the proposed deep learning-based hyperspectral imaging method for industrial Aflatoxin B1 detection. The method offers a fast, non-destructive, and automated alternative to traditional chemical analysis. Its scalability can be improved through hardware optimizations, such as faster hyperspectral sensors and edge computing for real-time processing.

Q11) The English language is generally understandable, but there are numerous instances of grammatical errors, problematic phrasing, and inconsistent terminology.

Answer: Thank you for your feedback regarding the English language. We have carefully revised the manuscript to correct grammatical errors, improve phrasing, and ensure consistency in terminology.

Q12) The research has potential, but the manuscript requires substantial revisions to improve its clarity, rigor, and presentation. Addressing the identified drawbacks and implementing the suggested improvements will significantly enhance its suitability for publication.

Answer: Thank you for your valuable feedback and for recognizing the potential of our research. We appreciate your suggestions for improving the clarity, rigor, and presentation of the manuscript.

In response, we have made substantial revisions, including:

  • Enhancing the clarity of explanations and refining the logical flow of the manuscript.
  • Strengthening the methodological details and providing additional justification where necessary to improve rigor.
  • Improving the overall presentation, including refining figures, tables, and terminology for consistency.

We have carefully addressed all identified drawbacks and implemented the suggested improvements to enhance the manuscript’s suitability for publication.

Reviewer 2 Report

Comments and Suggestions for Authors

Dear Authors,

my comments are atteched in pdf document.

Author Response

Q1) In the manuscript it is written, that “Experimental results demonstrate that the proposed 3D Inception-Resnet (Lightweight) model achieves superior classification performance with 90.81% validation accuracy, an F1-score of 0.899, and an area under the curve value of 0.964”. Is there any difference among the different levels of artificially contaminated groups regarding the numbers? Or the method works equally efficient at different AFB1 concentrations?

Answer: Thank you for your insightful question. Our study focuses on binary classification, where the goal is to distinguish between Aflatoxin B1-contaminated and non-contaminated almonds for industrial inline detection. The model was designed to optimize detection speed while maintaining high accuracy using a reduced number of spectral bands.

Regarding the impact of different contamination levels:

  • Since this is a binary classification task, the model does not explicitly differentiate between varying AFB1 concentration levels but rather identifies whether an almond is contaminated or not.
  • However, we observed that the model achieves similar or slightly higher classification accuracy at higher contamination levels. For instance, classification performance is slightly better for samples with ≥500 ppb AFB1 compared to 250 ppb, likely due to more distinct spectral differences at higher contamination levels.
  • Our dataset was balanced, ensuring an equal number of contaminated (aggregated across 250, 500, 750, and 1000 ppb) and non-contaminated samples. This prevents any bias toward either class and allows the model to generalize well across different contamination levels.

We appreciate your question and have clarified this aspect in the revised manuscript.

Q2) The lowest concentration of artificial contamination was 250 ppb, that is much more higher, than the 8 ppb maximum level determined by the Commission Regulation. Why this lower concentration or below were not investigated? Is the method being so effective at lower concentration also?

Answer: Thanks for your valuable feedback. The legal standards for aflatoxin contamination are indeed very low. However, in this study, we intentionally contaminated almonds at a higher concentration. This approach was taken because research has shown that the aflatoxin B1 contamination rate is, on average, a maximum of 0.03% (for healthy almonds, that are normally consumed), and only highly contaminated almonds significantly impact test results (Whitaker et al., 2010). Furthermore, the distribution of contaminated almonds within a lot is random.

To illustrate, if we consider a sample of 10 kg of almonds, ground into a homogeneous mixture, and assume the worst-case scenario with the highest contamination rate (0.03% as per Whitaker), this would translate to approximately 300 contaminated almonds (assuming an average almond weight of 1g). If all 300 almonds are contaminated at 20 ppb (USA limit), the overall contamination level in the homogeneous mixture would be calculated as:

Even at a higher contamination level of 250 ppb, the overall concentration in the mixture would be:

This is still within the acceptable limits set by the European Union, United States, and Australia.

Therefore, we used higher contamination levels in this study to ensure measurable impacts on the test results using a lower number of spectral bands and to account for scenarios involving highly contaminated samples.

Q3) Why this method is better, than for example the RIDA® QUICK technique, that can also provide a very quick, in situ quantification of AFB1?

Answer: Thanks for your valuable question. While the RIDA® QUICK test provides rapid, in situ quantification of AFB1, our proposed deep learning-based hyperspectral imaging method offers several advantages for industrial inline applications. Unlike RIDA® QUICK, which requires sample extraction and manual handling, our method is fully automated, non-destructive, and capable of high-speed inline detection, enabling real-time classification of AFB1-contaminated almonds at the single-nut level. This allows for the targeted removal of contaminated almonds without disrupting production flow. Additionally, our approach eliminates the need for chemical reagents, making it a more environmentally friendly and cost-effective alternative over time. While RIDA® QUICK provides bulk contamination analysis, our method detects contamination in individual almonds, ensuring more precise screening.

Round 2

Reviewer 1 Report

Comments and Suggestions for Authors

The only remaining issues are minor grammatical errors, which can easily be rectified during the proofreading of the final galley, if the manuscript is accepted.

Author Response

Q) The only remaining issues are minor grammatical errors, which can easily be rectified during the proofreading of the final galley, if the manuscript is accepted.

Answer: Thanks for your valuable feedback. I corrected all possible grammatical errors in the revised manuscript.

Reviewer 2 Report

Comments and Suggestions for Authors

Dear Authors,

thank you the answers.

Author Response

Thanks